# Possible Prophylactic Effects of Sulforaphane on LPS-Induced Recognition Memory Impairment Mediated by Regulating Oxidative Stress and Neuroinflammatory Proteins in the Prefrontal Cortex Region of the Brain

**DOI:** 10.3390/biomedicines12051107

**Published:** 2024-05-16

**Authors:** Noor Ahmed Alzahrani, Khulud Abdullah Bahaidrah, Rasha A. Mansouri, Rahaf Saeed Aldhahri, Gamal S. Abd El-Aziz, Badrah S. Alghamdi

**Affiliations:** 1Department of Biochemistry, Faculty of Sciences, King Abdulaziz University, Jeddah 23218, Saudi Arabia; kahmedbahaidrah@stu.kau.edu.sa (K.A.B.); amansouri@kau.edu.sa (R.A.M.); raldhahri0020@stu.kau.edu.sa (R.S.A.); 2Department of Biochemistry, Faculty of Sciences, University of Jeddah, Jeddah 23218, Saudi Arabia; 3Department of Clinical Anatomy, Faculty of Medicine, King Abdulaziz University, Jeddah 22252, Saudi Arabia; dr_gamal_said@yahoo.com; 4Department of Physiology, Neuroscience Unit, Faculty of Medicine, King Abdulaziz University, Jeddah 21589, Saudi Arabia; 5Neuroscience and Geroscience Research Unit, King Fahd Medical Research Center, King Abdulaziz University, Jeddah 21589, Saudi Arabia

**Keywords:** Alzheimer’s disease, neuroinflammation, lipopolysaccharides, recognition memory, sulforaphane

## Abstract

Background: Alzheimer’s disease (AD) presents a significant global health concern, characterized by neurodegeneration and cognitive decline. Neuroinflammation is a crucial factor in AD development and progression, yet effective pharmacotherapy remains elusive. Sulforaphane (SFN), derived from cruciferous vegetables and mainly from broccoli, has shown a promising effect via in vitro and in vivo studies as a potential treatment for AD. This study aims to investigate the possible prophylactic mechanisms of SFN against prefrontal cortex (PFC)-related recognition memory impairment induced by lipopolysaccharide (LPS) administration. Methodology: Thirty-six Swiss (SWR/J) mice weighing 18–25 g were divided into three groups (*n* = 12 per group): a control group (vehicle), an LPS group (0.75 mg/kg of LPS), and an LPS + SFN group (25 mg/kg of SFN). The total duration of the study was 3 weeks, during which mice underwent treatments for the initial 2 weeks, with daily monitoring of body weight and temperature. Behavioral assessments via novel object recognition (NOR) and temporal order recognition (TOR) tasks were conducted in the final week of the study. Inflammatory markers (IL-6 and TNF), antioxidant enzymes (SOD, GSH, and CAT), and pro-oxidant (MDA) level, in addition to acetylcholine esterase (AChE) activity and active (caspase-3) and phosphorylated (AMPK) levels, were evaluated. Further, PFC neuronal degeneration, Aβ content, and microglial activation were also examined using H&E, Congo red staining, and Iba1 immunohistochemistry, respectively. Results: SFN pretreatment significantly improved recognition memory performance during the NOR and TOR tests. Moreover, SFN was protected from neuroinflammation and oxidative stress as well as neurodegeneration, Aβ accumulation, and microglial hyperactivity. Conclusion: The obtained results suggested that SFN has a potential protective property to mitigate the behavioral and biochemical impairments induced by chronic LPS administration and suggested to be via an AMPK/caspase-3-dependent manner.

## 1. Introduction

Globally, 60–90% of dementia cases are caused by Alzheimer’s disease. Alzheimer’s disease (AD) is a neurological disorder characterized by an irreversible and gradual deterioration of cognitive functions, including learning and memory [1]. Individuals with AD also exhibit non-cognitive impairments such as mood disturbances, anxiety, appetite turbulences, and aggressive behavior [2]. The main histopathological hallmarks of AD are the extracellular deposits of amyloid beta (Aβ) plaques, intracellular neurofibrillary tangles (NFTs), and synaptic degeneration [1]. It has been revealed that Aβ deposition is initially constructed 10–20 years before the appearance of the cognitive decline [3].

Neuroinflammation is an immune response within the central nervous system (CNS) that could involve chronic stimulation of neuroglial cells and cytokine production [4]. In an AD brain, most reactive microglia are associated with Aβ plaques [5]. Moreover, research has demonstrated a proportional increase in both the number and size of reactive microglia with the size of these plaques [6]. Thus far, the link between neuroinflammation and Aβ pathogenesis in AD has been investigated in several studies [7,8,9].

Lipopolysaccharide (LPS) is an endotoxin found naturally in the outer membrane of Gram-negative bacteria [10]. It is broadly known as a stable and potent molecule and is able to induce a central inflammatory response [11,12]. The systemic administration of LPS to animals, even at low doses or through a single injection, is a commonly used experimental model to investigate the effect of neuroinflammation on behavior, cognition, and neurochemistry [13]. Animals exposed to LPS display cognitive impairment and a range of behavioral alterations, including reduced appetite, decreased movement, weight loss, impaired exploratory behavior, anxiety, and disrupted sleep patterns [14,15,16]. Indeed, most of these symptoms are assumed to be broadly similar to humans’ clinically related symptoms of neurodegenerative disease [17]. Moreover, a study by J.W. Lee and colleagues in 2008 suggested that administering LPS injections to mice for seven consecutive days could potentially boost the production and aggregation of Aβ in the brain cortex [9]. In addition, the neuroinflammation induced by peripheral LPS altered BBB integrity, thus disturbing Aβ peptide influx and efflux (clearance) from the brain, consequently accelerating pharmacological treatments [7]. In this context, studies using LPS as an experimental model have been able to provide a link between neuroinflammation and related events that are observed in humans with AD [7].

To date, there is no cure for AD dementia. In addition, no prescribed treatment is believed to stop the disease progression [18]. Indeed, researchers are focusing their efforts on interventions aimed at preventing disease progression in presymptomatic individuals, with increasing attention being paid to nutraceuticals as an alternative or complementary therapy [19]. From a wide variety of organosulfur compounds, sulforaphane (SFN) has received a significant amount of scientific consideration for its potent antioxidant, anti-inflammatory, and antiapoptotic biological actions [20,21,22,23]. For instance, Lee et al. (2014) reported that SFN alleviated scopolamine-induced memory impairment in mice [24]. A more recent investigation by Park et al. (2021) further corroborated these findings by demonstrating that SFN could reverse both short-term working memory and long-term spatial memory deficits caused by scopolamine administration [25].

On the other hand, a study by Shirai et al. (2015) showed that pretreatment with SFN (30 mg/kg) for 10 days had a prophylactic effect on phencyclidine-induced recognition memory deficits [26]. Furthermore, Subedi (2019) aimed to determine the molecular mechanism underlying the neuroprotective effect of SFN-enriched broccoli sprouts on scopolamine-induced cognitive impairment. The results of this study revealed that the oral administration of broccoli sprout 30 min before scopolamine administration improved the recognition ability during NORT in the scopolamine group [27].

However, despite the numerous investigations to understand SFN function and activity, little is known regarding its neuroprotective effects on recognition memory impairments. Therefore, this study aims to investigate the possible protective effects of SFN and underline the effects of its mechanisms on the brain, cognitive function, and behavior in LPS-induced recognition memory impairment in mice.

## 2. Materials and Methods

### 2.1. Animals

For this research, 36 adult male Swiss mice (SWR/J) within the weight range of 18 to 25 g were sourced from the animal facility at the King Fahd Medical Research Center (KFMRC), King Abdulaziz University, Jeddah, Saudi Arabia. The mice were accommodated in cages, with 3 mice per cage, and a controlled environment at a temperature of 23 ± 2 °C and a humidity level of 65% was maintained. The mice were kept in a standard 12/12 h light/dark cycle and were provided with unrestricted access to both water and standard food. All experiments were performed according to the guidelines of the biomedical ethics research committee (Reference No. 603-20) at King Abdulaziz University and followed the rules and regulations of the Animal Care and Use Committee at the KFMR which comply with guidelines of “System of Ethics of Research on Living Creatures” prepared by King Abdulaziz City for Science and Technology and were approved by the Royal Decree No. M/59 dated 24 August 2010.

### 2.2. Drug Preparations

#### 2.2.1. Preparation of LPS for the Induction of Recognition Memory Impairment in Mice

LPS (*E. coli* O111:B4) was acquired from In vivo Gen in France. A stock solution of LPS, with a concentration of 5 mg/mL, was prepared by dissolving 5 mg of LPS powder in 1 mL of endotoxin-free water. This solution was then divided into 100 μL aliquots and stored at −20 °C [10], with all injections being freshly prepared at the dose of 0.75 mg/kg each day in the morning and diluted from the stock solution to achieve the desired concentration (0.1 mL/10 g of body weight, administered i.p.).

#### 2.2.2. Sulforaphane Preparation

Sulforaphane was dissolved in 3% dimethyl sulfoxide (DMSO). The DMSO was prepared by diluting 3 mL of DMSO in 7 mL of normal saline to make a total volume of 10 mL. The SFN was freshly prepared and injected in the morning. Moreover, the average weight of each cage was recorded daily; therefore, the dose was prepared based on the calculation of reagent mass, the total volume of solution for injection, and mouse weight. Furthermore, SFN injections were prepared at the dose of 25 mg/kg of mouse weight with an injection volume of 0.2 mL/kg. The SFN solution was prepared in a bottle and covered with foil to protect it from light.

### 2.3. Experimental Design

The mice were assigned randomly to three groups, with each group consisting of 12 animals. (I) The control group received i.p. saline vehicle (0.9%); (II) the LPS group received an i.p. LPS dose of 0.75 mg/kg; and (III) the LPS +SFN group received an i.p. SFN dose of 25 mg/kg. The study spanned a total of 3 weeks. In the first week, the mice in the control and LPS groups were injected daily with i.p. normal saline, while the SFN group was injected with i.p. SFN (25 mg/kg). In the second week, recognition memory impairment induction was carried out, where mice in each group received 2 i.p. injections daily: the control group received saline and DMSO 3%; the LPS group received LPS 0.75 mg/kg and DMSO 3%; and the SFN group received SFN 25 mg/kg and LPS 0.75 mg/kg (Table 1). All treatments were administered between 11 a.m. and 12 p.m. Finally, in the last week, all study groups were subjected to behavioral tasks in order to test different types of recognition memory (Figure 1). On day 21, mice were sacrificed, and their brains were isolated and collected and stored in −80 °C for further investigation (Figure 1).

### 2.4. Assessment of Body Weight and Temperature

The daily measurement of body weight and temperature during the LPS administration was conducted before the injection to monitor the mice’s overall well-being. Their body temperatures were assessed using a DT-8826 non-contact infrared thermometer manufactured by SCC Inc. in Gampaha, Sri Lanka, following the prescribed device technique and protocol. Moreover, the percentage of body weight gain was calculated on day 14 in comparison with the initial weight using this equation: {[(mice weight at the end of LPS administration − mice initial weight)/initial weight] × 100}.(1)

### 2.5. Brain Extraction

The mice were anesthetized via the inhalation of isoflurane (Baxter, Baxter Health care of Puerto Rico, Deerfield, IL, USA) until the absence of reflex in tail or paw and immediately decapitated using large scissors. The complete brains were isolated and fixed in 10% paraformaldehyde fixation for further histological examination or stored at −80 °C for the biochemical analysis, which included ELISA and colorimetric investigations.

### 2.6. Behavioral Tests

#### 2.6.1. Open Field Test (OFT)

The OFT is a standard method utilized to assess locomotor activity. Locomotor assessment through OFT reflects the locomotive function in rodents, which is an important step in validating the results of novel object recognition memory tasks. Thus, OFT was conducted to check the normal locomotor activity of the mice [28,29]. The mice were habituated to the examiner for 1 week and to the test room for 30 min before the tests. Each mouse was gently placed in the center of the rectangular field (45 × 45 × 34 cm) facing the wall and was allowed to explore the arena freely for 10 min. Afterwards, they were returned to the home cage, then the arena was cleaned with 10% ethanol after each mouse to avoid odor cues. Sitting in the corners of the arena for the entire duration of the trial was considered an exclusion criterion. Animal movements were tracked and recorded using a digital camera and were analyzed via the EthoVision XT8A system (Noldus Information Technology, Wageningen, the Netherlands). Locomotor activity was evaluated via total distance moved (TDM) and velocity.

#### 2.6.2. Novel Object Recognition Test (NORT)

The NORT is a fast and efficient, widely applied method to evaluate recognition memory functions in mice [30]. Here, it was used to investigate the prophylactic effect of SFN on short-term memory. The experimental procedure was conducted following the methodology outlined in the study by Labban et al. (2021) [29]. The task was completed over two days: day 1 was dedicated to the habituation phase, and day 2 was for the test phase. During the habituation phase, the mice were allowed to move freely within the arena for 10 min. After 24 h, the mice entered the familiarization phase, during which each mouse encountered two identical objects for 3 min. Following a 10 min retention interval in their home cage, the mice were reintroduced to the arena for the test phase. During this phase, one of the familiar objects was replaced with a new object that possessed distinct characteristics in terms of shape, color, and texture (Figure 2A). Moreover, the arena was cleaned with 10% ethanol after each trial to remove odor cues. In addition, to guarantee that each mouse had an equitable opportunity to explore both objects during both phases, the frequency of sniffing of each object was assessed as follows:Frequency of sniffing (%) = (novel or familiar object frequency of sniffing/total frequency of sniffing of the two objects) × 100.(2)

Exploratory behavior was defined as pointing the nose at an object less than 2 cm away and touching it with the nose without turning or sitting on it. The EthoVision XT8A video tracking system was used to automatically track the mice’s movements and to record the time spent investigating each object (Figure 2B). Regarding the indexes of memory, the preference index (PI), discrimination index (DI), and recognition index (RI) were calculated. The calculation of PI and DI demonstrates the ability of the mice to differentiate between novel and familiar objects. In addition, RI was determined as the main index of retention. The memory indexes were calculated as follows:DI = (time spent exploring the novel object − time spent exploring the familiar object/total exploration time)(3)
RI = (time spent exploring the novel object/total objects exploration)(4)

PI = the percentage of time spent at each object divided by the time spent exploring both objects, i.e., time spent exploring the familiar or novel object/total exploration time × 100 (%) [30].

#### 2.6.3. Temporal Order Recognition Test (TORT)

Temporal order recognition memory is a higher-order memory task, defined as the ability to remember the sequence of past experiences to plan for future goals and actions [31]. In this study, a temporal order recognition test (TORT) was performed to examine the PFC-related temporal order recognition (TOR) memory function [32]. Generally, the TORT is based on an animal’s capacity to discriminate how recently they explored each object [33]. Moreover, this test comprised two sample phases (S1–S2) that were divided by an inter-sample interval (ISI) of 1 h. S1 entailed exposure to a set of two copies of the same object for 10 min followed by an ISI. In S2, the mice were presented with another two identical objects that differed from the previous set in S1, also for 10 min. To provide a retention delay after the sample phases, the test phase was carried out 1 h after S2. In the test phase, the mice were presented with two objects: one from S1 and the other from S2 (Figure 3A). Mice with intact TOR should prefer the exploration of the object presented earlier in the sequence (the object from S1) [33]. The EthoVision XT8A video tracking system was used to automatically record the time spent investigating each object during the test (Figure 3B). TOR memory performance was expressed via the DI and the exploration time (ET) [32].

### 2.7. Histological Examination

#### 2.7.1. Hematoxylin–Eosin (H&E) Staining

Following 48 h of immersion in a 10% neutral buffer formalin (NBF) fixation solution, sagittal brain slices were prepared to a thickness of 2 mm and processed to obtain paraffin blocks, which were subsequently sectioned by manual microtome into 4 µm sections, which were mounted on glass slides and stained with hematoxylin and eosin (H&E) according to Bancroft and Layton et al. (2013) [34].

#### 2.7.2. Congo Red Staining

For Congo red staining, paraffin blocks were also sectioned into 5–10 µm sections and stained according to the kit protocol (Dako: Code Number AR161). The staining procedure is based on using sodium chloride to reduce the background electrochemical staining in addition to enhancing the hydrogen bonds between Congo red dye and amyloid, which then demonstrates a noticeable dichroism under a polarized light microscope.

#### 2.7.3. Ionized Calcium Binding Adaptor Molecule 1(Iba1) Immunostaining

The detection procedure was conducted according to the kit instructions (ABCAM, catalog no. ab108539, 1:300).

### 2.8. Colorimetric Assessments

#### 2.8.1. Protein Extraction

Samples were prepared according to the manufacturing instructions of the kits. First, with a razor blade, the PFC was removed from the brain, weighed, and manually homogenized on ice using lysis buffer by a disposable homogenizing pestle according to the proportion of tissue: weight (1 g:10 mL). Homogenized PFC samples were then centrifuged at 8000× *g* for 10 min at 4 °C to remove insoluble materials. Finally, the supernatant was divided into 100 μL aliquots and stored at −80 °C.

#### 2.8.2. Determination of Superoxide Dismutase (SOD) Activity

The SOD activity in the PFC was measured using the detection kit (BC0170) from Solarbio (Beijing Solarbio Science & Technology Co., Ltd., Beijing, China) according to the manufacturer’s instructions which were derived from the method described by Spitz et al. (1989) [35]. Assessing superoxide dismutase (SOD) activity typically involves the measure of the enzyme’s ability to catalyze the dismutation of superoxide anions (O^2−^) into oxygen (O_2_) and hydrogen peroxide (H_2_O_2_). In this method, SOD inhibits the reduction of NBT by scavenging of superoxide anions (O^2−^), resulting in decreased formation of formazan. This reduction reaction leads to the formation of a blue-colored formazan product that exhibits its peak absorbance at 560 nm. The intensity of the blue color is indicative of the level of superoxide radicals present in the sample and can be used to assess the activity of SOD, where a more blue color of the reaction solution indicates lower SOD activity, while lighter blue colors indicate higher activity. One unit of enzyme activity was characterized as the quantity of enzyme necessary to catalyze a 50% inhibition within the reaction system. The inhibition percentage should be in the range of 30% to 70%. Higher SOD activity indicates greater antioxidant capacity, while lower activity may suggest increased oxidative stress.

#### 2.8.3. Determination of Catalase (CAT) Activity

The CAT activity in the PFC was determined using the detection kit (BC0200) from Solarbio (Beijing Solarbio Science & Technology Co., Ltd., Beijing, China) according to the manufacturer’s instructions and based on the method described in Johansson et al. (1988) [36]. Briefly, this procedure relies on CAT’s capability to decompose hydrogen peroxide (H_2_O_2_) into water and oxygen, thereby leading to a reduction in reagent absorbance. The absorbance of each sample was recorded both initially and postreaction at 240 nm. The rate of absorbance change is directly proportional to CAT activity. The result was expressed in (U/g of sample weight) in which one enzyme unit is defined as the quantity of enzyme that facilitates the decomposition of 1 μmol of H_2_O_2_ within the reaction system per minute per gram of tissue sample.

#### 2.8.4. Determination of Malondialdehyde (MDA) Level

The MDA in PFC was determined using a detection kit (BC0020) from Solarbio (Beijing Solarbio Science & Technology Co., Ltd., Beijing, China) according to the manufacturer’s instructions and drawing upon the methodology outlined in Draper and Hadley et al. (1990) [37]. Under conditions of high temperature and acidity, MDA reacts with thiobarbituric acid (TBA) to create a brownish-red compound which can be detected at a maximum wavelength of 532 nm. However, the presence of soluble sugars can interfere with this detection process due to their reaction with TBA, resulting in absorption wavelengths at both 450 nm and 532 nm. Thus, the MDA content was calculated by the difference between the absorbance at 532 nm, 450 nm, and 600 nm. Higher MDA levels indicate increased lipid peroxidation and oxidative stress in the sample. The MDA content was expressed in (nmol/g of sample weight).

#### 2.8.5. Determination of Reduced Glutathione (GSH) Content

The level of reduced glutathione (GSH) in PFC was determined using a detection kit (BC1170) from Solarbio (Beijing Solarbio Science & Technology Co., Ltd., Beijing, China) according to the manufacturer’s instructions and based on the method detailed by Owens and Belcher et al. (1965) [38]. Glutathione is a sulfhydryl group (-SH)-containing compound. Moreover, it can react with 5,5′-dithiobis-(2-nitrobenzoic acid) to produce 2-nitro-5-mercaptobenzoic acid and glutathione disulfide to produce 2-nitro-5-mercaptobenzoic acid which has a maximum absorption at 412 nm. The increase in absorbance over time corresponds to the formation of a yellow-colored product, which is proportional to the concentration of GSH in the sample. GSH concentration in the sample was determined by assessing the alteration in absorbance over time, utilizing a standard curve established with known concentrations of GSH standards. Higher GSH levels indicate increased antioxidant capacity and potential protection against oxidative stress. The GSH content was expressed in (μg/g of sample weight).

#### 2.8.6. Determination of Acetylcholine Esterase (AchE) Activity

The AChE activity was determined using a detection kit (BC2020) from Solarbio (Beijing Solarbio Science & Technology Co., Ltd., Beijing, China) according to the manufacturer’s instructions. The method was developed by Ellman in the early 1960s [39]. Briefly, the determination of AChE activity involves assessing the enzyme’s ability to hydrolyze acetylcholine (ACh) to generate choline which reacts with 2-nitrobenzoic acid.

The (5,5′-dithiobis-(2-nitrobenzoic acid) (DTNB) was used to form 5-mercapto nitrobenzoic acid (TNB). Increased absorbance rate at 412 nm over time corresponds to TNB formation. One unit of enzyme activity is defined as the amount of enzyme that catalyzes the generation of 1 nmol TNB in the reaction system per minute for every gram of sample. The enzyme activity was expressed as U/g of fresh weight. 

All colorimetric assays were performed in triplicate. Samples absorbance readings were obtained via a spectrophotometer (BioTek Instruments, Inc., Winooski, VT, USA) at KFMRC, Jeddah, Saudi Arabia.

### 2.9. Enzyme-Linked Immunosorbent Assay (ELISA)

#### 2.9.1. Sample Preparation

Samples were prepared according to the kit’s instructions. Initially, the PFC was chopped into 1–2 mm pieces using a disposable homogenizing pestle and homogenized in lysate solution with an ultrasonicate tissue homogenizer (BioLogics Inc., Manassas, VA, USA) on ice. The homogenized samples then underwent two freeze–thaw cycles to fully extract the protein. After, the Bradford method was used to determine the concentration of total protein in each sample.

#### 2.9.2. Assessment of TNF-α and IL-6 Levels

Brain TNF-α and IL-6 levels were determined via sandwich ELISA kits obtained from MyBioSource, San Diego, CA, USA (TNF-α: MBS825075 and IL-6: MBS824868) according to the kit instructions. In brief, the test procedure involved adding sample/standard and blank to a microplate, incubating for 90 min, and washing the plate. Subsequently biotin-labeled detection antibodies, streptavidin-HRP working solution, and 3,3′,5,5′-Tetramethylbenzidine (TMB) substrate solution were added, followed by stopping the reaction. Optical absorbance of each well was read within 30 min.

#### 2.9.3. Assessment of Phosphorylated AMPK and Caspase-3 Levels

The phosphorylated AMPK level (p-AMPK) was determined via a mouse competitive ELISA kit obtained from MyBioSource, San Diego, CA, USA (competitive ELISA, pAMPK (MBS7251953) and caspase-3 (MBS7210856)) according to the kit instructions. Briefly, samples, standards, and blanks were added to designated wells. PBS was added to the blank control well. The balance solution was added to the sample wells followed by mixing. Conjugate was then added to each well and incubated at 37 °C for 60 min. After incubation, the wells were washed multiple times with diluted wash buffer solution. Substrates A and B were added and incubated for 20 min before the reaction was stopped with a stop solution. Optical absorbance was measured at 450 nm within 30 min. All ELISAs were performed in triplicate. Sample absorbance readings were obtained via a microplate reader (BioTek Instruments, Inc., Winooski, VT, USA) at KFMRC, Jeddah, Saudi Arabia.

### 2.10. Statistical Analysis

All data were expressed as mean ± standard error of the mean and were statistically analyzed using GraphPad Prism 8.3.8 (GraphPad Software Inc., San Diego, CA, USA). One-way analysis of variance (ANOVA) followed by Tukey’s post hoc test was used to compare differences between the groups, including weight gain (%), TDM, velocity, DI, RI, PI, total exploration time, CAT activity, SOD activity, GSH content, MDA content, AChE activity, TNF-α level, IL-6 level, p-AMPK level, and active caspase-3 level. However, regarding the frequency of sniffing, body weight, and body temperature, a two-way ANOVA followed by Tukey’s post hoc test was used. The differences between the groups were considered statistically significant when the *p*-value was <0.05.

## 3. Results

### 3.1. The Chronic Administration of LPS Had a Temporary Reduction Effect on Mouse Body Weight and Did Not Affect Body Temperature

During the 14-day drug administration, the two-way repeated-measures ANOVA for body weight indicated a significant difference only on days 9 and 10 (*F* (26, 429) = 11.84, *p* < 0.0001). After, from day 11 the body weight reduction startedto revert to the baseline. Further, weight gain (%) showed no statistically significant difference between the LPS and the control groups, nor between the LPS and the LPS + SFN groups (control, *p* = 0.9949, and LPS + SFN, *p* = 0.3918) (Figure 4B). In addition, the two-way repeated-measures ANOVA of body temperature showed no significant difference for days × groups (*F* (26, 308) = 1.088, *p* = 0.3530) (Figure 4C).

### 3.2. The Chronic Administration of LPS Did Not Affect Locomotor Activity during OFT

Compared to the LPS group, total distance traveled showed no statistically significant difference in the control and LPS + SFN groups (control, *p* = 0.3355, and LPS + SFN, *p* = 0.1924) (Figure 5A). The average velocity per trial was assessed and compared between groups as a locomotion indicator. There was no significant difference in the velocity between LPS and other groups (control, *p* = 0.4162; and LPS + SFN, *p* = 0.9943) (Figure 5B). The representative tracking pathways of each group are shown in Figure 5C.

### 3.3. Prophylactic Effect of SFN on Novel Object Recognition (“What” Memory)

#### 3.3.1. Sulforaphane Protects against Exploratory Behavior Impairment Induced by the Chronic Administration of LPS

The total exploration time spent around the objects and frequency of sniffing (%) during the familiarization and test phase were calculated in order to detect whether exploratory behavior was affected by the LPS or SFN injections [29]. During the familiarization phase, there was no significant difference in the frequency of sniffing (%) of each object (familiar vs. familiar) among the groups (control, *p* > 0.9999; LPS, *p* > 0.9999; LPS + SFN, *p* = 0.8383) (Figure 6A). In the test phase, the LPS group showed no significant difference in frequency of sniffing (%) between the two objects (familiar vs. novel, *p* = 0.1801). On the other hand, frequency of sniffing (%) was significantly greater for the novel object than for the familiar object in the control and LPS + SFN groups (control, *p* = 0.0001; LPS + SFN, *p* < 0.0001). Moreover, the calculation of the cumulative time spent with both objects (familiar vs. familiar) during the familiarization phase revealed no significant difference in all groups (control, *p* > 0.9999; LPS, *p* > 0.9999; LPS + SFN, *p* > 0.9999) (Figure 6B). Conversely, the test phase group comparisons showed that the mice in the LPS group spent an equal amount of time exploring the two objects (familiar vs. novel, *p* > 0.9999), during which the mice in the control and LPS + SFN groups had a longer novel object exploration time (control, *p* = 0.0008; LPS + SFN, *p* = 0.0037) (Table 2).

#### 3.3.2. Sulforaphane Protects against Short-Term Recognition Memory Impairment Induced by the Chronic Administration of LPS

In this study, representative traces for all groups during familiarization and test phases are shown in Figure 7. The preference index (%) was calculated during the test phase (Figure 8A). The LPS group showed no significant difference in this index regarding novelty preference between the two objects (familiar vs. novel, *p* > 0.9999). In contrast, there was a significant increase in this percentage in both the control and the LPS + SFN groups (control, *p* < 0.0001; LPS + SFN mg/kg, *p* < 0.0001). In comparison to the control group, the DI was significantly decreased in the LPS group (*p* < 0.0001), while the LPS + SFN group showed a significant difference in DI (*p <* 0.0001) compared to the LPS group (Figure 8B). As shown in Figure 8C, there was a significant decrease in RI in the LPS group compared to the mice in the control group (control, *p* < 0.0001). Instead, the LPS + SFN group showed a significant difference compared to the LPS group (*p* < 0.0001).

### 3.4. Sulforaphane Protects against Temporal Order Recognition Memory Impairment Induced by the Chronic Administration of LPS

The representative tracking pathways of each group are shown in Figure 9. Compared to the control group, the DI was significantly decreased in the LPS group (*p* < 0.0001). However, the LPS + SFN group showed a significant difference compared to the LPS group (*p* = 0.0018) (Figure 10A). Calculation of RI for the LPS group showed a significant difference compared to the control group (old vs. recent, *p* < 0.0044), while the mice in the LPS + SFN showed a significant difference compared to the LPS group (old vs. recent, *p* < 0.9999) (Figure 10B).

### 3.5. Sulforaphane Protects against Neuronal Death in the PFC Induced by the Chronic Administration of LPS

In the control group (Figure 11A,B), the examination of H&E sections of the PFC revealed a normal structure in the form of six successive layers from the outside to the inside: the outer molecular layer (ML), external granular layer (EGL), external pyramidal layer (EPL), internal granular layer (IGL), internal pyramidal layer (IPL), and polymorphic layer (PL). The higher magnification of these layers showed that the outer molecular layer under the pia mater mainly contained the dendrites of underlying cells with some neuroglial cells. The other layers were formed mostly of mixed pyramidal and granular cells. The pyramidal cells appeared triangular with large vesicular nuclei, while the granular cells appeared rounded with large vesicular nuclei. The background or neuropil appeared pinkish and contained some blood vessels and neuroglial cells. In contrast, the examination of the H&E-stained sections of the PFC from the LPS group (Figure 11C,D) exhibited disturbed layers and the cellular distribution of neurons with increased darker neurons as compared with the control group. Most of the pyramidal and granular cells appeared condensed and deeply stained. There were also more neuroglial cells and congested capillaries. On the other hand, the examination of H&E sections of the prefrontal cortex from the treated group (Figure 11E,F) showed a restoration of the normal arrangement and cellular distribution of neurons when compared to the LPS group. Most of the pyramidal and granular cells appeared normal in shape with a few cells that were shrunken and deeply stained. There were also fewer neuroglial cells and fewer congested capillaries.

### 3.6. Sulforaphane Protects against PFC-Aβ Deposition Induced by the Chronic Administration of LPS

Congo-red-stained sections of the control group showed no detection of any amyloid plaque deposition in the different regions of the PFC (Figure 12A). However, the LPS group sections showed a moderate deposition of amyloid plaques (with eosinophilic core surrounded by glial cells) in the upper layers of the PFC (Figure 12B). Moreover, the sections of the LPS + SFN group showed no detection of any amyloid plaque deposition in the different regions of the PFC (Figure 12C).

### 3.7. Sulforaphane Protects against PFC Microglial Activation Induced by the Chronic Administration of LPS

The IHC study of microglia using Iba-1 antibodies in the PFC of the control group (Figure 13A) revealed mild to moderate expression of microglia in all layers. High magnification showed that the microglia displayed small cell bodies with fine and elongated processes projecting out from their cell bodies. On the other hand, the LPS group sections revealed the intense expression of microglia in all layers. High magnification showed that the microglia displayed larger cell bodies and shorter and coarser cytoplasmic processes (Figure 13B). Further, the sections of the LPS + SFN group (Figure 13C) revealed the moderate expression of microglia in all layers. High magnification revealed that the microglia displayed small cell bodies with fine and elongated processes projecting out from their cell bodies, similar to those of the control group.

### 3.8. Sulforaphane Protects against PFC-OS Induced by the Chronic the Administration of LPS

Compared to the control group, the MDA levels in the mice PFC were remarkably induced in the LPS group (*p* < 0.0001) (Figure 14A). Moreover, the activity of CAT and SOD, in addition to that of the GSH levels, was reduced (*p* = 0.0026, *p* = 0.0155, and *p* = 0.0042) (Figure 14B–D). On the other hand, LPS + SFN results showed a significant reduction in MDA levels (*p* = 0.0001), an increase in CAT and SOD activity (*p* = 0.0064 and *p* = 0.0317), and an increase in GSH levels (*p* = 0.0190).

### 3.9. SFN Protects against Increased TNF-α and IL-6 Levels Induced by the Chronic Administration of LPS

As shown in Figure 15A, the level of TNF-α was significantly elevated in the LPS group compared to the controls (*p* < 0.0001). The mice treated with SFN (25 mg/kg) showed significantly lower levels of TNF-α compared to the LPS group (*p* < 0.0001). Moreover, the levels of IL-6 were significantly higher in the LPS group compared to the control (*p* < 0.0001). On the other hand, LPS + SFN group demonstrated a significantly lower level of IL-6 compared to the LPS group (*p* < 0.0001) (Figure 15B).

### 3.10. SFN Protects against Increased AChE Activity Induced by the Chronic Administration of LPS

As shown in Figure 16, AChE activity was significantly increased in the LPS group compared to the control group (*p* = 0.0091), while mice in the LPS + SFN group revealed a significant reduction in AChE activity compared to the LPS group (*p* = 0.0001).

### 3.11. SFN Protects against Increased Cleaved Caspase-3 Levels Induced by the Chronic Administration of LPS

There were statistically significant increases in cleaved caspase-3 levels compared to the control group (*p* = 0.0003). In contrast, cleaved caspase-3 levels were significantly lower in the LPS + SFN group compared with the LPS group (*p* < 0.0001) (Figure 17).

### 3.12. SFN Protects against Decreased Phosphorylated AMPK Levels Induced by the Chronic Administration of LPS

The LPS group showed a significantly decreased phosphorylated (active) AMPK level compared to the control group (*p* = 0.0023). In contrast, the LPS + SFN group indicated a significant increase in the active AMPK levels compared with the LPS group (*p* = 0.0073) (Figure 18).

## 4. Discussion

### 4.1. Effect of the Chronic Administration of LPS on Mice’s General Health

Body weight gain (%) and temperature showed no significant differences between all of the study groups. On the other hand, the daily body weight records showed that mice administered LPS only exhibited a significant body weight reduction on days 9 and 10 of the experiment. However, from day 11 until the end of the administration period, the earlier decrease in body weight was normalized with no significant differences among groups.

In addition, the analysis of the TDM and velocity showed no significant differences between any of the groups, indicating that treatment with LPS or SFN did not affect mouse locomotor activity. In agreement with our results, the studies of Kahn et al. (2012) and Jung et al. (2023) reported that sickness behavior (defined by reduced locomotion and body weight loss) induced by the chronic peripheral administration of LPS was reversible [40,41]. The results showed that mice were able to develop a tolerance to the sickness behavior while maintaining cognitive deficits indicated by the MWMT and NORT. Thus, the authors proposed that the weight gain detected from the fourth day of the LPS injections is a result of the sickness behavior inhibition.

### 4.2. Prophylactic Effect of SFN against Recognition Memory Impairment Induced by Chronic Administration of LPS

During the NORT, mice with a damaged PFC appeared to have no preference for either object and explored both of them equally, whereas mice with an intact PFC preferred to explore the novel object over the familiar one [42]. Our NORT results demonstrated exploratory behavior and recognition memory impairments in LPS mice. The calculation of the frequency of sniffing (%) and the cumulative exploration time showed that the LPS group had no significant exploratory preference for either object, as previously reported in the literature [28]. In contrast, the estimation of the same parameters in the LPS + SFN group exhibited a significant preference for the novel object over the familiar object.

In addition, PI, DI, and RI were also determined as memory indicators. Compared to the control group, the mice in the LPS group failed to recognize the novel object and showed a significant reduction in all indexes, as previously reported in the literature [43,44,45]. Instead, the mice in the LPS + SFN group were significantly able to discriminate between novel and familiar objects. Regarding the literature, only a few studies discussed the effect of SFN on recognition memory. However, all of these studies yielded consistent results: SFN was shown to improve recognition memory impairment. A study on scopolamine-induced neuroinflammation by Subedi et al. (2019) demonstrated that 2 weeks of SFN-enriched broccoli sprout treatment significantly increased the ability of mice to recognize a novel object, as indicated by the RI [27].

Furthermore, to obtain an overall understanding of the prophylactic effects of SFN on recognition memory performance, the TORT was also utilized. According to Barker et al. (2007), PFC lesions impaired TOR memory in the TORT, as animals with a damaged PFC did not remember the order of previously explored objects [33]. Interestingly, no previous study has investigated the effect of chronic LPS or SFN administration on TOR memory. Here, our findings proved, for the first time, that chronic LPS administration can cause TOR memory impairment, as indicated by the decreased DI. The DI demonstrated that the mice failed to distinguish the more recent familiar object (S2 object) from the older familiar object (S1 object). Furthermore, calculations of the total exploration time showed no preference for either object by the LPS group as they explored both similarly. However, the mice in the LPS + SFN group were able to differentiate between both objects based on the order of presentation to a remarkable extent.

### 4.3. Prophylactic Effect of SFN against PFC Neurodegeneration Induced by Chronic Administration of LPS

The histological examination by H&E staining was performed to link neurodegeneration in the PFC with recognition memory impairments. Compared to the control group, the brain sections from the LPS group showed notable cortical neurodegeneration, as reported previously in Alzahrani et al. (2022) [28], whereas sections from the LPS + SFN group revealed that pretreatment with SFN prevented neuronal abnormal morphological alterations and degeneration [46].

### 4.4. Prophylactic Effect of SFN Administration against PFC-Aβ Fibrillar Deposition Induced by Chronic Administration of LPS

Additionally, Congo red staining was used to investigate the PFC-Aβ fibrillar deposition [47]. Aβ deposition has been identified as one of the major contributing factors to gradual neuronal degeneration and synaptic loss, which ultimately leads to cognitive impairments in AD [48]. In agreement with our results, several animal studies have shown a significant link between LPS administration and Aβ aggregations [9,49]. In this study, Congo red staining exhibited a remarkable increase in Aβ deposition within the LPS group.

On the other hand, the results from the LPS + SFN group showed decreased Aβ deposition in the PFC sections. Our results are consistent with the findings of Hou et al. (2018), who explored the effects of SFN on spatial memory. The study confirmed that the prophylactic effect of SFN after 4 months of administration was able to inhibit Aβ oligomer production and peptide processing in PS1V97L transgenic mice [50]. Additionally, a study of an AD transgenic mouse model demonstrated that oral gavage with SFN (10 or 50 mg/kg) was also able to improve Aβ clearance [51].

### 4.5. Prophylactic Effect of SFN against PFC Microglial Activation Induced by Chronic Administration of LPS

A great deal of previous research has focused on the effect of LPS on glial cells, and it is well established that LPS is able to induce microglial overactivity [52,53]. In contrast, a marked decrease in active microglia was seen in mice given both LPS and SFN. This is in accordance with previous reports, demonstrating the potential of SFN to prevent microglial activation [27,54,55].

### 4.6. Prophylactic Effect of SFN against Antioxidant Defense Defect Induced by Chronic Administration of LPS

Here, the antioxidant defense status was estimated to assess SFN’s prophylactic effects on LPS-induced OS. Oxidative stress and excessive ROS release are believed to be early events in AD progression [56]. In fact, one of the major pathological mechanisms of LPS is to disturb cellular antioxidant defenses and instigate the release of mitochondrial ROS [57]. In addition, increased MDA levels were also linked with LPS exposure [58]. In this study, chronic LPS administration was able to induce OS, as indicated by increased MDA and reduced GSH, as well as decreased CAT and SOD activity. These results broadly correlate with several studies [59,60,61]. On the other hand, the results from the LPS + SFN group showed that the prophylactic effects of SFN mitigated all these changes to a remarkable extent.

These results showed that the neuroprotective effect of SFN can be attributed to its antioxidant potential. Despite this, data on the neuroprotective efficacy of SFN against oxidative damage are limited. However, an interesting finding was revealed by Sun et al. (2013) [62], who studied the therapeutic effect of SFN on lead-induced spatial memory impairment. The study found that there was no statistical difference in SOD activity in the SFN group compared to the AD group. These results could indicate that the beneficial effect of SFN on SOD activity may possibly be more efficient as a protective rather than a therapeutic method. Further, our results support the evidence of other preclinical studies that Nrf2 activation and the upregulation of its target genes are the main neuroprotective mechanisms of SFN [63,64,65].

### 4.7. Prophylactic Effect of SFN against Higher Concentrations of Proinflammatory Cytokines Induced by Chronic Administration of LPS

To further identify the possible prophylactic mechanisms of SFN on recognition memory impairments, the levels of inflammatory markers (TNF-α and IL-6) were evaluated. The ability of LPS to increase the secretion and activation of cytokines was suggested to be its main mechanism to stimulate Aβ deposition and aggregation and may also delay amyloid plaque clearance [66]. The estimation of TNF-α and IL-6 levels of the LPS group revealed a significant increase, as has been extensively indicated in different studies [7,13,67]. In contrast, the prophylactic effect of SFN significantly decreased IL-6 and TNF-α levels, which underlines the finding that the neuroprotective effect of SFN may also depend on its anti-inflammatory action.

However, only two studies have investigated the anti-inflammatory activity of SFN in an LPS mouse model [68,69]. Indeed, our findings support the work of the other studies. In male C57BL/6J mice, i.p. injections of SFN (20 mg/kg) with LPS (0.25 mg/kg) for 7 consecutive days reversed TNF-α and IL-1β upregulation in the hippocampus [68]. Further, the second study proved that treatment with SFN modulated elevated brain IL-6 and TNF-α levels induced by a single injection of LPS (0.50 mg/kg) [69].

### 4.8. Prophylactic Effect of SFN against AChE Overactivation Induced by Chronic Administration of LPS

The prophylactic effect of SFN on memory-related parameters was demonstrated by measuring AChE activity status. AChE is considered to be a key regulatory enzyme in the cholinergic nervous system [70]. Increased AChE activity makes the cell more susceptible to apoptosis and serves as an intrinsic regulator of inflammation [71,72]. In the present investigation, LPS administration resulted in the enhancement of AChE activity, which correlates with several previous studies [73,74]. Conversely, the SFN was considerably able to prevent this increase. Nevertheless, a limited number of studies have investigated the prophylactic effects of SFN on the cholinergic system. Generally, our results match those that support the evidence of SFN’s potential effects on the improvement of the cholinergic system, thus improving learning and memory [24,75]. For instance, in mouse models of scopolamine-induced memory impairment, the administration of SFN (10 or 50 mg/kg) to C57BL/6 mice via oral gavage for 2 weeks significantly reduced AChE activity in the frontal cortex [24]. Likewise, Subedi et al. (2019) used the same mouse model and showed that treatment with SFN-enriched broccoli sprouts for 2 weeks resulted in the significant reparation of recognition memory decline. Moreover, researchers attribute this memory improvement to the significant reduction in AChE activation [27].

### 4.9. Prophylactic Effect of SFN against Increased Caspase-3 Induced by Chronic Administration of LPS

We extended our interest to exploring the antiapoptotic mechanism by which the prophylactic effect of SFN reduces neurodegeneration through the assessment of active caspase-3 levels in the PFC. Caspase-3 is the final executioner in the apoptotic process [76]. Moreover, it is principally synthesized as a procaspase that must cleaved to be active [77].

As reported in several earlier studies, the current study revealed that upregulation of the cleaved caspase-3 levels was found in the LPS group samples [78,79]; specifically, findings from the LPS + SFN group showed decreased levels of the active caspase-3. This indicates that the prophylactic effects of SFN include an antiapoptotic mechanism, which can be mediated by caspase-3 inhibition.

In line with our findings, a previous study on the hypoxia mouse model demonstrated the cytoprotective effect of SFN (0.50 mg/kg) on hippocampus neurons and spatial memory due to the suppression of caspase-3-dependent neuronal apoptosis [80]. Similarly, the oral administration of SFN (25 mg/kg) once daily for 14 consecutive days was found to reduce caspase-3 expression in a diabetic rat model [81].

### 4.10. Prophylactic Effect of SFN on Reduced Phosphorylated AMPK Induced by Chronic Administration of LPS

Typically, it is well known that SFN is a potent Nrf2 stimulator, which is suggested to be the main target for its therapeutic and protective activity [63,65]. However, to gain more insight into the signaling pathways involved in its neuroprotective effect against recognition memory impairment, this study aimed to address a new mechanism. AMPK is a key regulator of cellular energy homeostasis [82]. Moreover, it is generally activated by site-specific phosphorylation, where the Thr172 residue of the α subunit is the main activation site [82]. Phosphorylated AMPK plays a vital role as a negative regulator of apoptosis, oxidative stress, and inflammation [83]. Active AMPK, directly or indirectly, inhibits several target proteins, including caspase-3 [84]. The beneficial effects of SFN have previously been linked with AMPK activation through in vivo and in vitro models of different diseases [83,85,86]. In fact, our study was the first to investigate AMPK activation as a possible neuroprotective mechanism of SFN in mice specifically in relation to neurobehavioral deficits. Here, the ELISA results showed a significant decrease in phosphorylated AMPK levels within the PFC of mice in the LPS group, which correlates with previous studies in lung and mammary epithelial cells [87,88,89]. By contrast, the results from the LPS + SFN group showed that the prophylactic effect of SFN increased the levels of phosphorylated AMPK in the PFC. In agreement with our results, The AMPK pathway activation was also identified as one of the protective mechanisms of SFN by an in vitro study on LPS-induced injury within intestinal epithelial cells [83].

Indeed, this study aimed to address a new protective mechanism of SFN against PFC-related recognition memory impairments. Regarding the literature, treatment with AMPK activators, such as resveratrol, AICAR, and metformin, has been reported to attenuate cognitive deficits, inflammatory responses, Aβ plaque, and cell apoptosis [90,91]. Various studies have investigated the effect of SFN on obesity [92], pancreatic cancer [93], and fatty liver disease [94], demonstrating that SFN can be considered an AMPK activator.

In this context, we can conclude that SFN increases AMPK phosphorylation, leading to the beneficial events that support neuronal survival. For instance, Li et al. (2018) demonstrated that the administration of an AMPK activator significantly enhanced the phosphorylated AMPK levels in the cerebral cortex [95]. Both in vivo and in vitro studies of microglia have identified the ability of AMPK activators to switch the microglial phenotype from M2 to M1 in an AMPK-activation-dependent manner [95]. Thus, the activation of AMPK also decreases the neurotoxic molecules secreted by M1 microglia, including proinflammatory cytokines and the ROS [96]. The activation of AMPK consequently suppresses the expression of proinflammatory downstream targets, such as TNF-α, IL-6, and IL-1β [96]. At the same time, AMPK enhances transcription of selected Nrf2 target genes to counteract OS and cell damage [97,98]. Previous research established that the AMPK regulatory functions of cellular energy homeostasis may have another direct protective effect on neurons [96]. Additionally, AMPK activation has been reported to mitigate neuronal death induced by Aβ deposition in the hippocampal and cortical regions [90]. Furthermore, AMPK activation was also suggested to prevent apoptotic cell death by decreased caspase-3 concentrations [99]. In this regard, we hypothesized, for the first time, that the activation of the AMPK/caspase-3 pathway is one of the SFN-specific neuroprotective mechanisms against recognition memory impairment.

### 4.11. Limitations

This study has several limitations, including the use of quantitative analysis for microglia and Congo red histological examinations. In addition, there is still a need for the further exploration of SFN’s effects on more downstream targets of AMPK, such as NF-κB isomers and FOXO.

## 5. Conclusions

In conclusion, this research project was designed to investigate the prophylactic effects of SFN on PFC-related recognition memory impairment induced by LPS and to elucidate the underlying mechanisms involved.

The results showed an improvement in mouse recognition memory performance during the NORT and the TORT with SFN administration. In addition, histological and IHC studies of the PFC showed that the prophylactic effects of SFN prevent PFC neurodegeneration and Aβ pathology, as well as microglial hyperactivation. Subsequently, we investigated the antioxidative and anti-inflammatory mechanisms by which SFN improves recognition memory impairment and PFC alterations.

Moreover, we investigated whether the prophylactic effects of SFN included the AMPK/caspase-3 pathway. Our results showed that pretreatment with SFN increases phosphorylated AMPK and decreases active caspase-3 levels. Since AMPK is well known for its antiapoptotic, anti-amyloidogenic, anti-inflammatory, and antioxidative activity, we suggested in this study that the protective effects of SFN could be mediated by AMPK/caspase-3 (Figure 19).

This finding provides a novel platform for exploring the mechanism underlying the neuroprotective effects of SFN on PFC-related recognition memory impairment and further contributes to the development of new therapeutic approaches for cognitive impairment caused by AD.

## Figures and Tables

**Figure 1 biomedicines-12-01107-f001:**
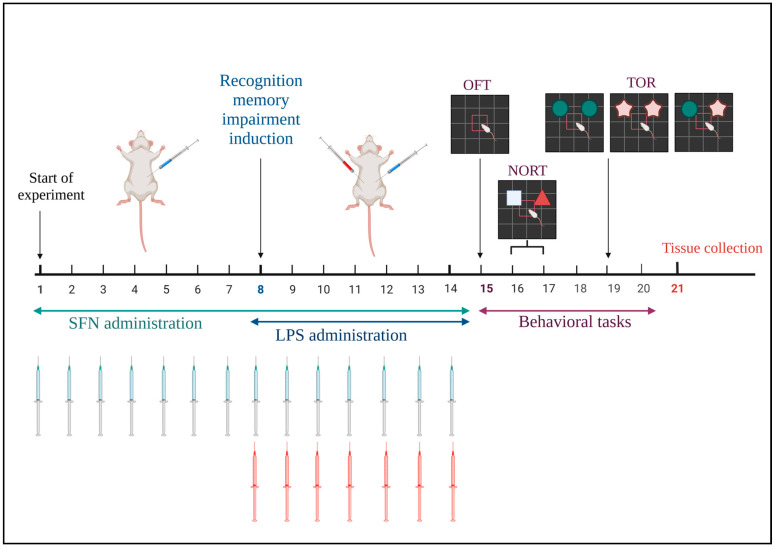
Timeline and design of the study experiment.

**Figure 2 biomedicines-12-01107-f002:**
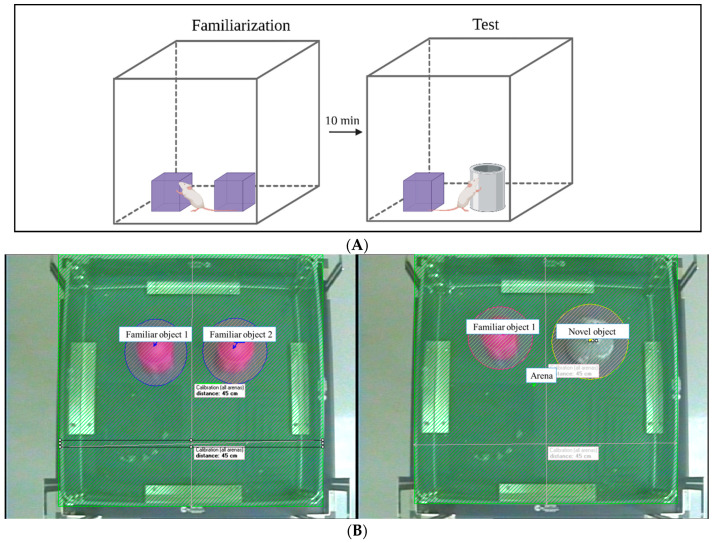
Novel object recognition test (NORT) protocol. (**A**) Illustration of the protocol. (**B**) Arena setting in EthoVision.

**Figure 3 biomedicines-12-01107-f003:**
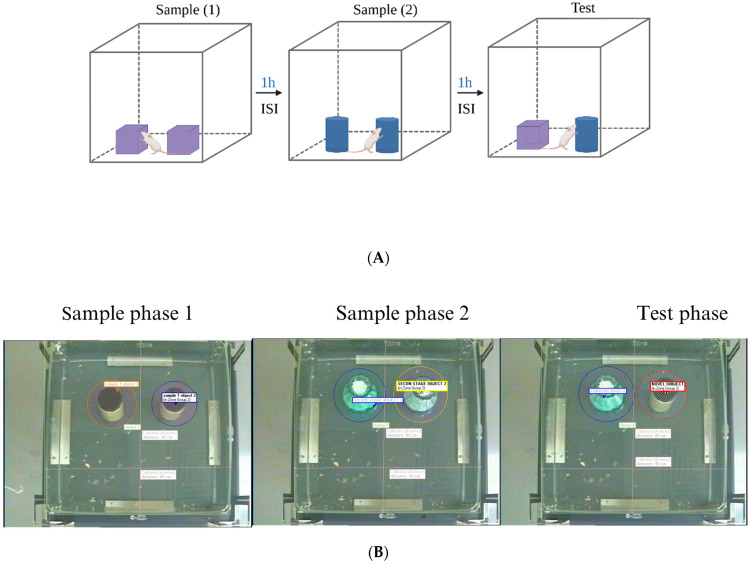
Temporal order recognition test (TORT) protocol. (**A**) Illustration of the protocol. (**B**) Arena setting in EthoVision.

**Figure 4 biomedicines-12-01107-f004:**
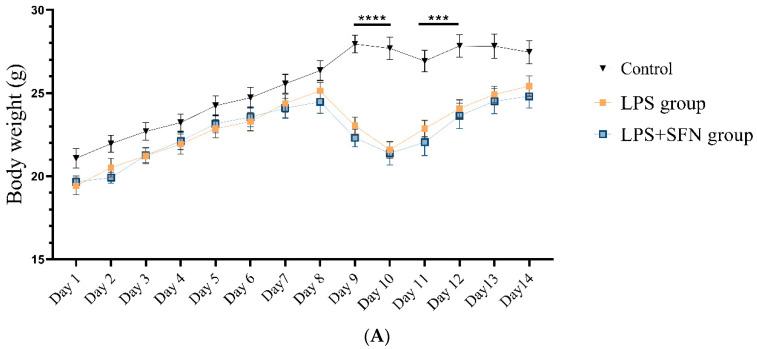
Effect of lipopolysaccharide (LPS) on (**A**) body weight (g), (**B**) body weight gain (%), and (**C**) body temperature. Body weight was measured daily during the 14 days of drug administration (**A**). The total body weight gain (%) at day 14 is calculated in (**B**). The temperature was measured daily during 14 days of drug administration (**C**). Data are presented as mean ± SEM (*n* = 12). A two-way ANOVA followed by the Bonferroni multiple comparison test was used for (**A**,**C**). One-way ANOVA followed by Tukey’s test was used for (**B**). *** *p* < 0.001, **** *p* < 0.0001. LPS, lipopolysaccharide; SFN, sulforaphane; SEM, standard error of the mean; ANOVA, analysis of variance; ns, non-significant.

**Figure 5 biomedicines-12-01107-f005:**
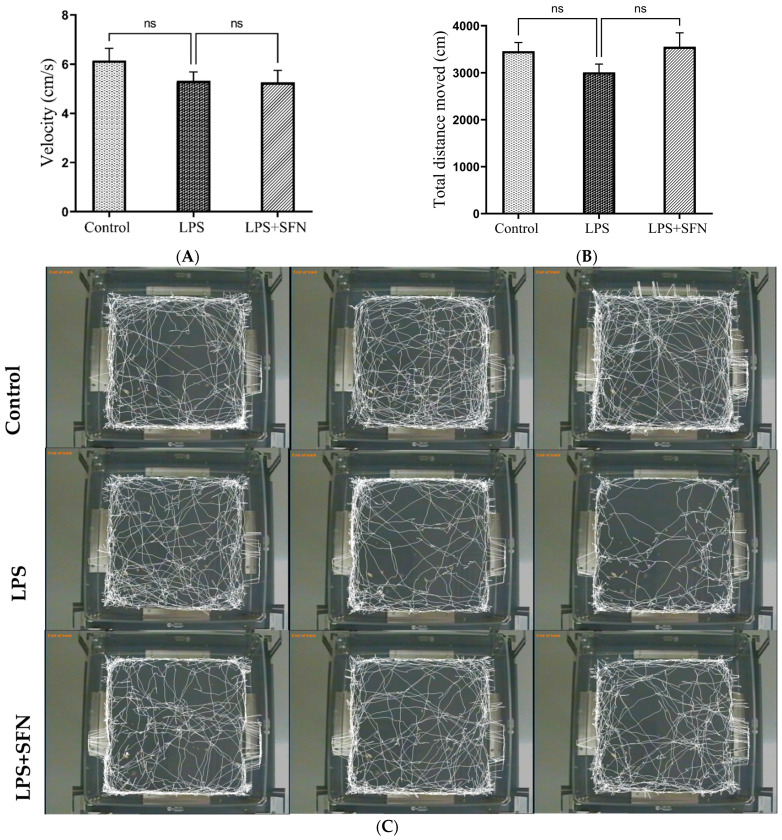
Locomotor activity assessment in the open field test (OFT). (**A**) There were no significant differences in velocity among the groups. (**B**) No significant differences were observed in the TDM among all groups. (**C**) Representative track pathways for the open field test. Data are presented as mean ± SEM (*n* = 12). One-way ANOVA followed by Tukey’s multiple comparisons test was used. LPS, lipopolysaccharide; SFN, sulforaphane; TDM, total distance moved; SEM, standard error of the mean; ANOVA, analysis of variance; ns, non-significant.

**Figure 6 biomedicines-12-01107-f006:**
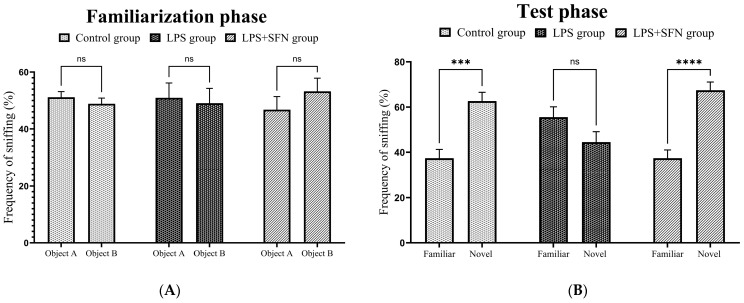
Effect of SFN on frequency of sniffing (%) in mice during novel object recognition test (NORT). (**A**) Among all groups, no significant differences were found during the familiarization stage in the frequency of sniffing (%) of the two identical objects (familiar 1 vs. familiar 2). (**B**) In the test phase, a significant difference was found in the frequency of sniffing (%) of each object (familiar vs. novel) in the control and LPS + SFN groups, but not the LPS group. Data are presented as mean ± SEM (*n* = 12). Two-way ANOVA followed by Tukey’s multiple comparisons test was used, *** *p* < 0.001, **** *p* < 0.0001. SFN, sulforaphane; LPS, lipopolysaccharide; NORT, novel object recognition test; ns, non-significant; SEM, standard error of the mean; ANOVA, analysis of variance.

**Figure 7 biomedicines-12-01107-f007:**
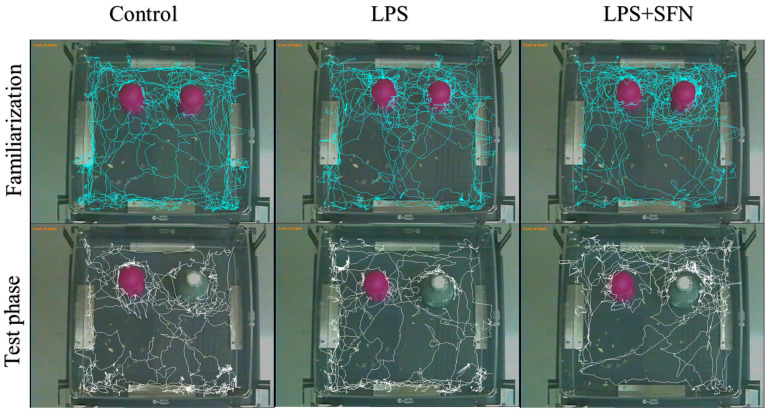
Representative track pathways for the novel object recognition test (NORT). SFN, sulforaphane; LPS; lipopolysaccharide.

**Figure 8 biomedicines-12-01107-f008:**
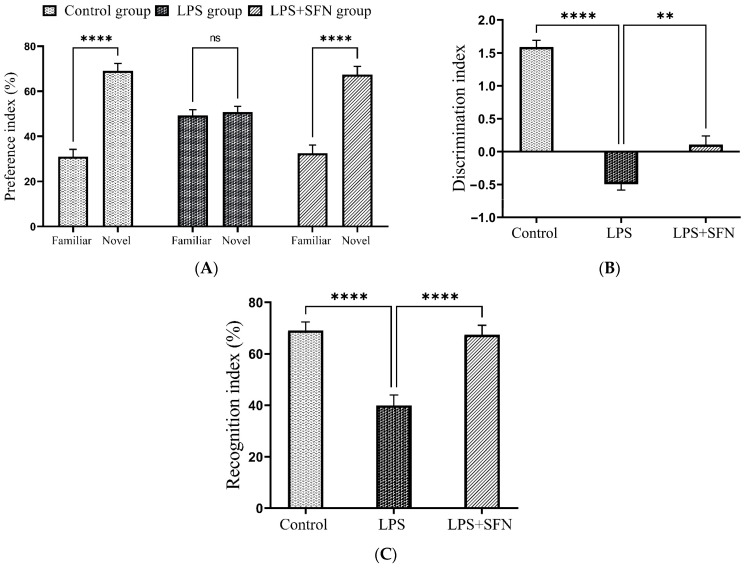
Effect of SFN on mice’s preference in the NORT. There was a significant difference in the PI (**A**), DI (**B**), and RI (**C**) in LPS vs. the control mice and LPS + SFN vs. the LPS mice. Data are presented as mean ± SEM (*n* = 12). One-way ANOVA followed by Tukey’s multiple comparisons test was used, ** *p* < 0.01, **** *p* < 0.0001. SFN, sulforaphane; LPS, lipopolysaccharide; NORT, novel object recognition test; PI, preference index; DI, discrimination index; RI, recognition index; ns, non-significant; SEM, standard error of the mean; ANOVA, analysis of variance.

**Figure 9 biomedicines-12-01107-f009:**
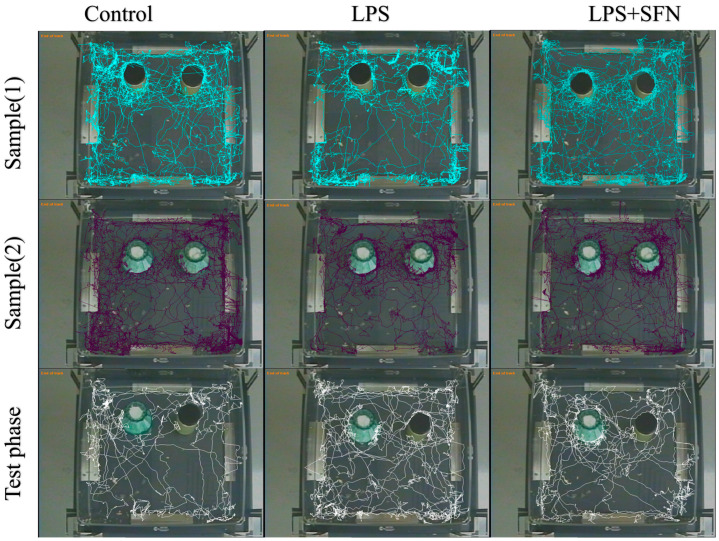
Representative track pathways of temporal order recognition test (TORT). SFN, sulforaphane; LPS, lipopolysaccharide; TORT, temporal order recognition test.

**Figure 10 biomedicines-12-01107-f010:**
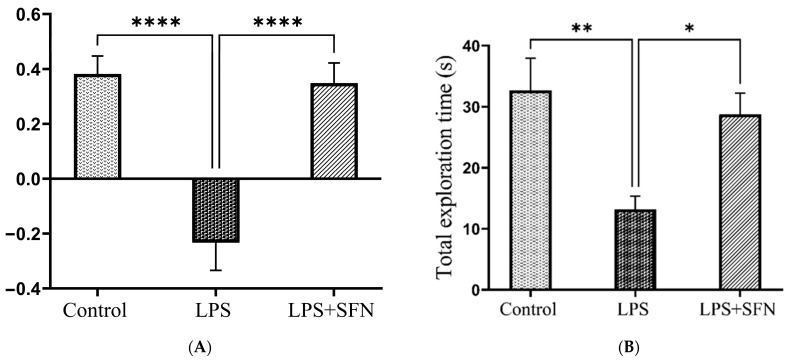
Effect of SFN on mice discrimination ability during temporal order recognition test (TORT). (**A**) A significant difference in DI was observed in LPS vs. the control mice and LPS + SFN vs. the LPS mice. (**B**) Calculation of the total exploration time reveals a significant difference in LPS vs. the control mice and LPS + SFN vs. the LPS mice. Data are presented as mean ± SEM (*n* = 12). One-way ANOVA followed by Tukey’s multiple comparisons test was used, * *p* < 0.1, ** *p* < 0.01, **** *p* < 0.0001. SFN, sulforaphane; LPS, lipopolysaccharide; TORT, temporal order recognition test; SEM, standard error of the mean; ANOVA, analysis of variance.

**Figure 11 biomedicines-12-01107-f011:**
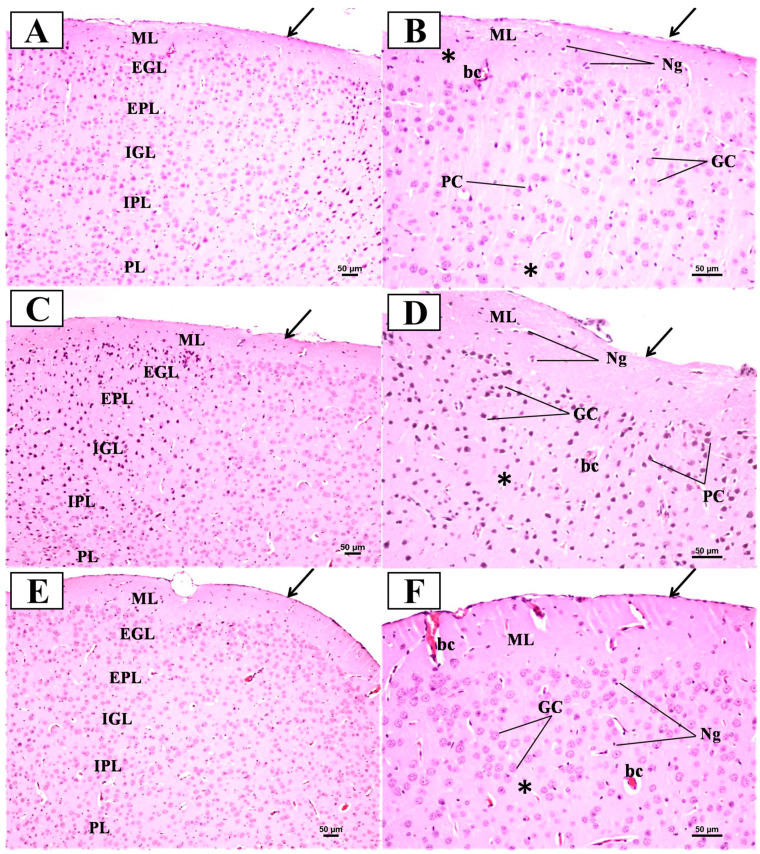
Representative photomicrographs of H&E-stained sections of the prefrontal cortex (PFC) from mice in the different groups. (**A**) Normally organized layers of the prefrontal cortex from superficial to deep: molecular (ML), external granular (EGL), external pyramidal (EPL), internal granular (IGL), internal pyramidal (IPL), and polymorphic (PL) layers. Arrow = regularly attached pia mater (H&E × 100). (**B**) A higher magnification of the upper part of the PFC displaying ML, EGL, and EPL reveals the appearance of granular cells (GC) and pyramidal cells (PC). Neuroglial cells (Ng) and blood capillaries (bc) within the acidophilic neuropil (*) can also be identified. (**C**) Irregularly attached pia mater (arrow) and the disturbed layers and cellular distribution of neurons of the PFC. (**D**) A higher magnification of the upper part of the PFC displaying ML, EGL, and EPL reveals more disturbed layers and cell distribution compared to the control group. Notice that most of the pyramidal and granular cells appear condensed and deeply stained. There were also more neuroglial cells (Ng) and congested capillaries (bc). (**E**) Regularly attached pia mater (arrow) and slightly arranged layers of the PFC that are bordered by white matter (wm). (**F**) A higher magnification of the upper part of the PFC displayed many apparently normal neurons, with a few cells appearing condensed together with apparently normal blood capillaries (bc) (H&E × 200). * refers to the background or neuropil that appeared pinkish and contained blood capillaries and neuroglial cells.

**Figure 12 biomedicines-12-01107-f012:**
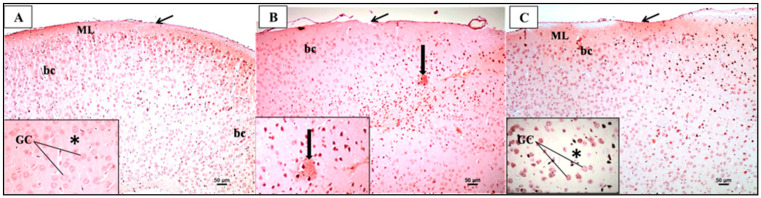
Representative photomicrographs of Congo-red-stained sections of the prefrontal cortex (PFC) from mice in the different groups. (**A**) Congo-red-stained sections with no detection of any amyloid plaque deposition. (**B**) Congo-red-stained sections with moderate deposition of amyloid plaques (with eosinophilic core surrounded by glial cells) on upper layers (thick arrow). (**C**) Congo-red-stained sections with no detection of any amyloid plaque deposition (Congo red ×100; inset ×200). * Refers to the background or neuropil that appeared pinkish and contained blood capillaries and neuroglial cells.

**Figure 13 biomedicines-12-01107-f013:**
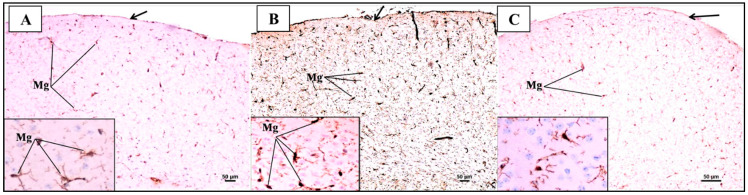
Representative photomicrographs of Iba1-stained sections of the prefrontal cortex (PFC) from mice in the different groups. (**A**) Immunohistochemical localization of microglia displayed mild to moderate expression in all layers that have small cell bodies with fine and elongated processes projecting out from their cell bodies. (**B**) Immunohistochemical localization of microglia displayed intense expression of activated microglia in all layers that have larger cell bodies with shorter and coarser cytoplasmic processes projecting out from their cell bodies. (**C**) Immunohistochemical localization of microglia displayed moderate expression in all layers that have small cell bodies with fine and elongated processes projecting out from their cell bodies (inset) (Iba-1 ×100; inset ×400).

**Figure 14 biomedicines-12-01107-f014:**
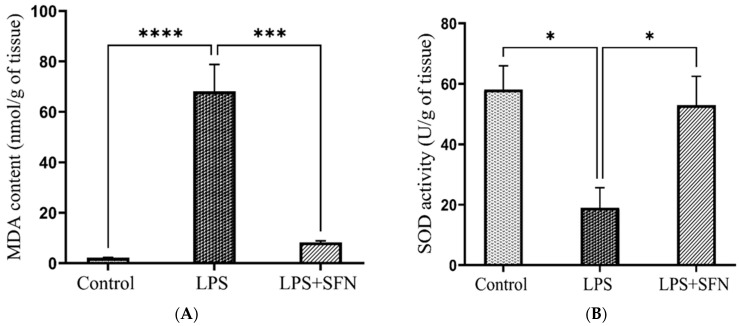
Effect of SFN on the antioxidant’s status in mice PFC, expressed as nmol/g of brain tissue. (**A**) MDA, (**B**) SOD, (**C**) CAT, and (**D**) GSH. A significant increase in oxidative stress was observed in LPS vs. the control mice. The LPS + SFN mice showed a notable reduction in oxidative stress compared to the LPS mice. Data are presented as mean ± SEM (*n* = 5). Two-way ANOVA followed by Tukey’s multiple comparisons test was used, * *p* < 0.1, ** *p* < 0.01, *** *p* < 0.001, **** *p* < 0.0001. SFN, sulforaphane; LPS, lipopolysaccharide; SEM, standard error of the mean; ANOVA, analysis of variance.

**Figure 15 biomedicines-12-01107-f015:**
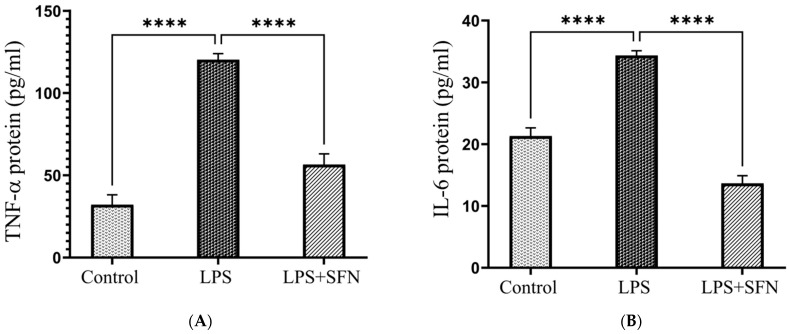
Effect of SFN on tumor necrosis factor α (TNF-α) and interleukin 6 (IL-6) levels in mouse PFC, expressed as pg/mL of brain tissue. A significant difference in (**A**) TNF-α and (**B**) IL-6 levels was found in LPS vs. control mice, and a significant difference in TNF-α and IL-6 levels was indicated in the LPS + SFN vs. the LPS mice. Data are presented as mean ± SEM (*n* = 6). Two-way ANOVA followed by Tukey’s multiple comparisons test was used, **** *p* < 0.0001. SFN, sulforaphane; LPS, lipopolysaccharide; SEM, standard error of the mean; ANOVA, analysis of variance.

**Figure 16 biomedicines-12-01107-f016:**
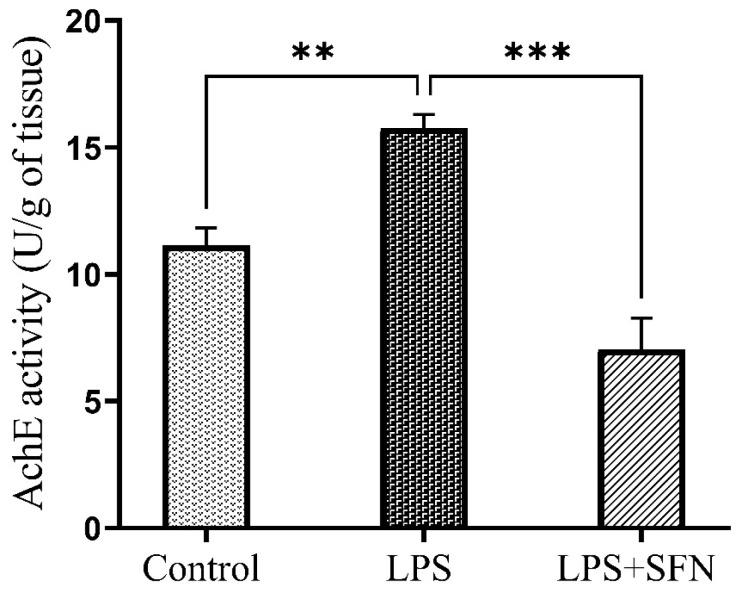
Effect of SFN on acetylcholinesterase (AChE) activity in mouse PFC, expressed as U/g of brain tissue. A significant difference in AChE activity was observed in LPS vs. the control mice. The AChE activity showed a notable difference in the LPS + SFN vs. the LPS mice. Data are presented as mean ± SEM (*n* = 5). Two-way ANOVA followed by Tukey’s multiple comparisons test was used, ** *p* < 0.01, *** *p* < 0.001. SFN, sulforaphane; LPS, lipopolysaccharide; SEM, standard error of the mean; ANOVA, analysis of variance.

**Figure 17 biomedicines-12-01107-f017:**
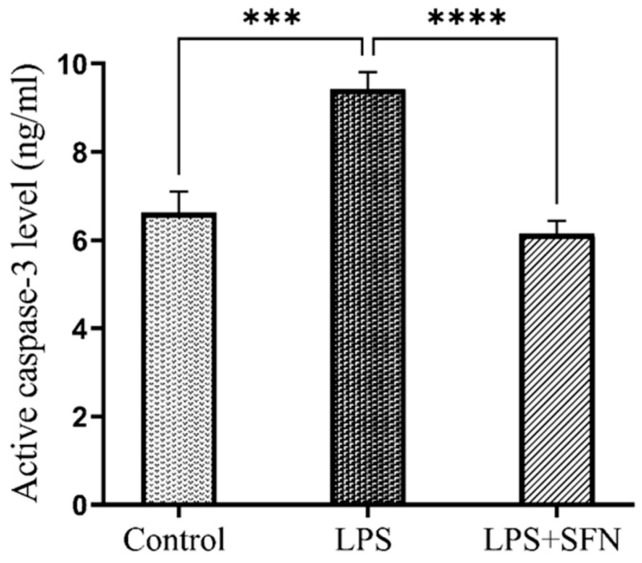
Effect of SFN on active caspase-3 level in mouse PFC expressed as ng/mL of brain tissue. A significant differance in active caspase-3 level was observed in LPS vs. the control mice. A significantly lower level of the active caspase-3 was found in the LPS + SFN vs. the LPS mice. Data are presented as mean ± SEM (*n* = 6). Two-way ANOVA followed by Tukey’s multiple comparisons test was used, *** *p* < 0.001, **** *p* < 0.0001. SFN, sulforaphane; LPS, lipopolysaccharide; SEM, standard error of the mean; ANOVA, analysis of variance.

**Figure 18 biomedicines-12-01107-f018:**
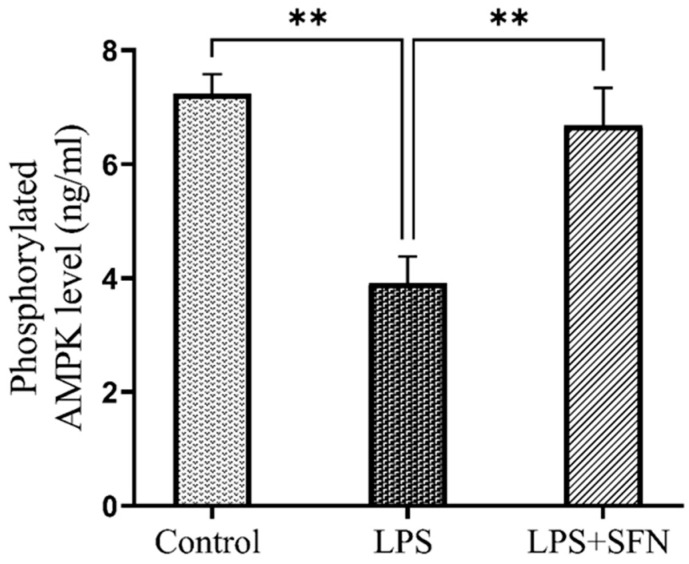
Effect of SFN on phosphorylated AMPK levels in mouse PFC expressed as ng/mL of brain tissue. A significant difference in phosphorylated AMPK levels was shown in the LPS vs. the control mice. A significant increase in phosphorylated AMPK levels was shown in the LPS + SFN vs. the LPS mice. Data are presented as mean ± SEM (*n* = 6). One-way ANOVA followed by Tukey’s multiple comparisons test was used, ** *p* < 0.01. SFN, sulforaphane; LPS, lipopolysaccharide; SEM, standard error of the mean; ANOVA, analysis of variance.

**Figure 19 biomedicines-12-01107-f019:**
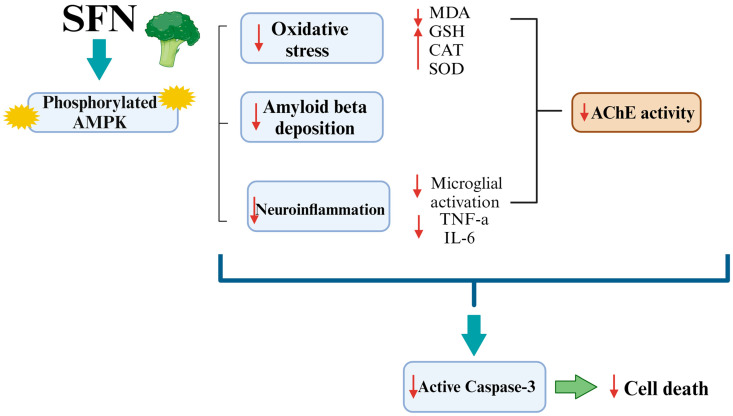
Diagrammatic illustration of the paper’s conclusion.

**Table 1 biomedicines-12-01107-t001:** Experimental design.

Group	Control	LPS	SFN
Week 1	i.p. Saline	i.p. Saline	i.p. SFN (25 mg/kg)
Week 2	i.p. Saline i.p. DMSO 3%	i.p. LPS (0.75 mg/kg) i.p. DMSO 3%	i.p. SFN (25 mg/kg) i.p. LPS (0.75 mg/kg)
Week 3	No injections were given. Behavioral tests were performed.		

**Table 2 biomedicines-12-01107-t002:** Total exploration time between the two objects during NORT.

Group	Familiarization Phase		Test Phase	
	Familiar	Familiar	Familiar	Novel
Control	15.69 ± 2.4	16.30 ± 1.98 ^ns^	13.06 ± 1.47	24.74 ± 1.50 ***
LPS	13.31 ± 2.09	15.08 ± 2.48 ^ns^	15.65 ± 2.41	16.89 ± 2.77 ^ns^
LPS + SFN	18.08 ± 2.37	15.79 ± 2.50 ^ns^	11.52 ± 1.75	21.80 ± 1.50 ****

Data are presented as mean ± SEM (*n* = 12). Two-way ANOVA followed by Tukey’s multiple comparisons test was used, *** *p* < 0.001, **** *p* < 0.0001. SFN, sulforaphane; LPS, lipopolysaccharide; NORT, novel object recognition test; ns, non-significant; SEM, standard error of the mean; ANOVA, analysis of variance.

## Data Availability

The original contributions presented in the study are included in the article.

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
