# Peer review of "Possible Prophylactic Effects of Sulforaphane on LPS-Induced Recognition Memory Impairment Mediated by Regulating Oxidative Stress and Neuroinflammatory Proteins in the Prefrontal Cortex Region of the Brain"

_biomedicines, 2024, doi:10.3390/biomedicines12051107_

Round 1
Reviewer 1 Report
Comments and Suggestions for Authors
The manuscript of Alzahrani and colleagues investigated the preventive effects of SFN against recognition memory impairment induced by lipopolysaccharides (LPS) administration in mice.
SFN improved recognition memory performance during behavioral tests, protected the cortex from neuroinflammation and oxidative stress, and reduced neurodegeneration, Aβ accumulation, and microglial hyperactivity. SFN can defend against a recognition memory impairment induced by LPS administration.
Unfortunately, the issue lacks novelty and lacks many recent references to the topic.
The results are not expressed very clearly An excellent analysis is made of the Nort but not of the biochemistry or the open field.
When it comes to researching motor skills, it's not uncommon to use the Nort habituation phase as an open field. However, this is not an appropriate approach when analyzing anxiety. In the case of anxiety research, it's recommended to use a separate arena and analyze the movements in the central area as well as the area near the center. Alternatively, you could create a special mask using the Noldus technology to measure explorations in the center area, permanence, time away from the center, and identify the comfort zone."This analysis is mandatory."
Regarding caspase 3, the authors mention a specific point in the results. "...LPS administration significantly increased cleaved (active) caspase-3 levels compared to the control group (P=0.0003). In contrast, SFN administration significantly reduced the cleaved caspase-3 levels in the LPS+SFN group compared with the LPS group(P<0.0001) (..."......... No figures were cited, and the sentence was incomplete.
Additionally, there were several typos and poorly concluded attempts to remove parts throughout the text.
The results obtained are indirect and there is no concrete evidence to support their functionality.
The text requires revision to enhance clarity and coherence.
Reviewer 2 Report
Comments and Suggestions for Authors
The authors studied the prophylactic effect of sulforaphane on LPS-induced cognitive deficits, neurodegeneration and changes of biochemical parameters related to oxidative stress, inflammation and apoptosis. Potentially, the study is interesting, but in the present form the manuscript cannot be considered for publication. In fact, before submission the authors should checked all parts of the manuscript and wrote the manuscript in accordance with the Instructions for authors. This version of the manuscript is far from its final form. With respect to the time of the reviewers, it is not correct to submit the manuscript:
a) that is not written in accordance with the Instructions for authors – Abstract (should without headings), Literature, Citations throughout the manuscript should be corrected
b) that needs English editing
c) in which some results are missing (caspase, AMPK)
d) that have completely inappropriate citations throughout the Discussion
e) that has false claims and content not relevant for the topic
Besides, antioxidative, anti-inflammatory, anti-apoptotic, cholinergic effects of SFN, as well as its ability to improve cognitive abilities have been decribed in the literature. In addition, the study is performed with only one dose. AD terminology is questionable, LPS-induced neuroinflammation is more appropriate.
Examples of bad English:
Line 52 – “is initially constructed 10–20 years before”
Line 74 – “thus disturbing Aβ peptide influx and efflux (clearance) from the brain which accelerating”
Line 77 – “its related events that observed in human with AD[10] .”
L174 – “mice entered the familiarization phase”
L277 – “until the conclusion of the drug administration period”
L567 “ was significantly increased mice ability to”
Examples of incorrect citations
Line 56, ref. 5 – ref. relevant for AD should be cited
In the discussion section, references cited are not in accordance with the corresponding text.
L630, L631, L655, L659 (ref. 75,76), L674 (ref. 82,83), L690 (ref. 61, 62),……… till the end of the Discussion section – ref. cited are not related to SFN or are not cited properly
Ref. 59-65 are not cited in the manuscript
Ref. 64 – the authors cited results on Sertoli cells (testicles) – not relevant to AD and cholinergic system
False claims
L577 – “Interestingly, no previous study has investigated the effect of chronic LPS or SFN administration on TOR memory.” - not correct
e.g. https://pubmed.ncbi.nlm.nih.gov/37926114/
https://www.scienceopen.com/document_file/1ce6e08c-66df-446f-b6e6-06b742b14be8/PubMedCentral/1ce6e08c-66df-446f-b6e6-06b742b14be8.pdf
L732 – “In fact, the present study appears to be the first study investigated AMPK activation as a neuroprotective mechanism of SFN.” – not correct
e.g. https://www.sciencedirect.com/science/article/abs/pii/S0306452214006496
Other:
Line 111 – “obtained from (Invivogen, France).” – brackets?
L116 – “(0.1 ml/10 g body weight, i.p.).” – dose should be indicated
Line 137 – -80, unit is missing
L157 – “Thus, OFT was conducted to ensure mice's normal locomotor activity[25] .” – not to ensure, to check
L472, L498 – titles 3.4. and 3.5 are not appropriate (the title should not be methodological)
Titles of the y axis in Figure 16 are not written correctly
L526 – “body weight gain (%) and temperature showed no significant differences between all the study groups (the control, LPS, and LPS+SFN) groups.” – simplification of the results, from the graph it is hard to believe that there were no differences in body weight gain
Why was the body weight not monitored for 21 days? Perhaps the time was needed for adverse effects to become visible.
L608 –“ To the best of our knowledge, no previous study has attempted to assess the anti-amyloidogenic effect of SFN using this method.” – In the Discussion section focus is on results, not on the method
L618 – “Different factors were reported to be involved to increase BACE1 activity, including the OS, free radicals, pro-inflammatory cytokine, and microglial hyperactivation [48]–[50].” – not relevant for the topic, should be deleted
L751 – “In a study of (HP-1c) which is a dual AMPK/Nrf2 activator, has been stated it as a powerful agent to reduce brain inflammation and OS [111].” - broadly relevant for the topic
The last Figure needs legend and title
The last Figure is too simplified and misleading, mechanisms are not clearly indicated, e.g. it looks that reduced cell death leads to reduced caspase-3 activity, it is not explained how is AMPK related to AChE activity in the text
Figure 2B is not clearly visible
Experimental procedures should be described in more details (e.g. 2.8, 2.9,…..)
Figure 4B – from the results presented in Figure 4A, it seems that LPS should be more similar to LPS+SFN than to control group; the data should be presented day by day, it is not clear what time point Figure 4B indicates
Figure 4 – marks of significance are missing in the legend
Figure 5 – A,B, C are missing in the legend, C should be explained
Figure 6 - To the untrained eye, it is hard to detect obvious differences between A, B and C, the authors should describe differences or indicate them with an arrow
Figure 8, 9 – representative track pathways (plural)
Figure 14, CAT activity – values on the y axis are not appropriate
Comments on the Quality of English Language
English editing is required.
Reviewer 3 Report
Comments and Suggestions for Authors
Dear authors, I enjoyed reading the manuscript entitled "Possible Prophylactic Effects of Sulforaphane on LPS-Induced Recognition Memory Impairment by Regulating Oxidative Stress and Neuro-Inflammatory Proteins in the Prefrontal Cortex Region of the Brain". The present paper brings numerous evidences of the beneficial effects induced by sulforane, in the case of neurodegenerative conditions, especially Alzheimer's Dementia
The manuscript is valuable because it is an original one, carried out in the preclinical area. The large number of images, tables, graphs, support the obtained results and offer the possibility to understand certain aspects more easily, making reading easier. A very large number of bibliographic references, some quite recent, are in agreement with the presented theme. The entire manuscript is well organized, systematized, presented in a very clear manner. Fairly extensive paraclinical investigations reinforce and support this study.
I just have a few questions and suggestions:
1. For this study, do the authors have Approval from the Ethics Committee?
2. What properties does sulforane have? Does it cross the blood-brain barrier? Is it lipophilic or? How does it reach the brain level?
3. What are the limitations of this study? Or maybe there are no limitations!
4. English could be slightly improved.
Comments on the Quality of English LanguageEnglish could be slightly improved.
Round 2
Reviewer 1 Report
Comments and Suggestions for Authors
The English revision of the manuscript was very helpful in making it more understandable.
However, the abstract should be rewritten, the methods extended, and the discussion revised.
Additionally, I appreciate that the authors provided me with the necessary data on behavior that I would like to include in the supplementary material to improve the quality of the paper. I also think it would be beneficial to discuss the authors' presence, as they connect two crucial aspects of anxiety and inflammation, with anxiety often preceding inflammation. Some authors suggest that anxiety can serve as a biomarker of inflammation (PMID: 32906843; PMID: 29173175).
The text lacks information about the number of words used for the different essays, such as ache or caspase. Whether the essays were made in triplicate, double, or single must be clear. Additionally, there is no description of how the bark is mechanically separated under the stereo microscope after cutting longitudinal slices. The text also refers to the localization of cortical areas but needs to provide more detail. To make the experiments repeatable by others and confirm the data, it is necessary to strengthen the methods with much more detail and provide elements such as the number of observations, animals, averages, and deviations.
Reviewer 2 Report
Comments and Suggestions for Authors
The authors did not correct all references that were used nonadequatelly. Just to mention few:
L752 - not cited properly
L755 – according to Barker, ref. 30, not correctly
L783 - [43]–[45] – not cited correctly
L830 - [64]- not cited correctly
…………
The authors should provide in track changes all corrections that were made to liturature and citations (starting from the original manuscript).
· Figure 6 - To the untrained eye, it is hard to detect obvious differences between A, B and C, the authors should describe differences or indicate them with an arrow
Response: Thank you for your comment. Figure 6 contains only A and B
I apologize. Please describe textually what differences are important to be seen by comparing Cont, LPS and LPS+SFN in Figures 8 and 9. A, B, C should be on the same page in the manuscript.
Figure 19, no sense, AcHE activity is not the major factor for reducing caspase-3 activity, this is what figure indicates.
Round 3
Reviewer 1 Report
Comments and Suggestions for Authors
The authors improved the manuscript in a significant way. Nevertheless, the problem of methods remains fundamental, which must contain appropriate references indicating how things were done and whether the method is accredited.
The methods needs to be revised and references need to be added for each technique used in the paper.
Author Response
Response: Thank you for recognizing the improvements made to the manuscript. We highly value your feedback regarding the methods section. We worked diligently to make this important by including appropriate references that outline the methodology used and considering accreditation where applicable. The method section is now completely revised. Your insights are valuable, and we're committed to addressing these changes by tracking them.
Reviewer 2 Report
Comments and Suggestions for Authors
After two rounds of revision some references are still not related to the text although this issue was mentioned two times before and authors were asked to carefuly check the references. Besides, references were neither written according to the instructions of the journal nor written in an uniform manner.
Here are some obvious examples only from the Discussion:
L780 - Subedi or Surbadi?
L816 - ref. 44 is not relevant
L840 - ref. 19 is not relevant
L980 - ref. 105 is not relevant
L988 - ref. 103 is not relevant
L995 - ref. 108 i s not relevant
L869 - ref. 12 is not related to SFN
L879 - PFC TNF-alpha and IL-6 levels, should be TNF-alpha and IL-6 levels in the PFC
In general, Discussion section should be shortened. Results are repeated in too many details.
Comments on the Quality of English Language
Minor editing
